# Collaborative Continuum Robots for Remote Engineering Operations

**DOI:** 10.3390/biomimetics8010004

**Published:** 2022-12-22

**Authors:** Nan Ma, Stephen Monk, David Cheneler

**Affiliations:** School of Engineering, Lancaster University, Lancaster LA1 4YW, UK

**Keywords:** continuum robot, dual-arm robot, collaborative operation, in situ operation

## Abstract

In situ repair and maintenance of high-value industrial equipment is critical if they are to maintain the ability to continue vital operations. Conventional single-arm continuum robots have been proven numerous times to be successful tools for use in repair operations. However, often more than one arm is needed to ensure successful operation within several scenarios; thus, the collaborative operation of multiple arms is required. Here, we present the design and operating principles of a dual-arm continuum robot system designed to perform critical tasks within industrial settings. Here, presented are the design principle of the robotic system, the optimization-based inverse kinematic calculation of the 6-DoF continuum arms, and the collaborative operation strategy. The collaborative principle and algorithms used have been evaluated by a set of experiments to demonstrate the ability of the system to perform in situ machining operations. With the developed prototype and controller, the average error between planned and real toolpaths can be within 2.5 mm.

## 1. Introduction

In recent years, continuum robots have increasingly been used to perform operations in confined spaces due to their inherent dexterity, compact structure, and ability to adapt to complex environments [1,2]. Generally, the length-to-diameter ratio of continuum robots is very high, enabling the delivery of end effectors deep into tight spaces with limited access [3,4]. Additionally, given the availability of compatible specialty end effectors (for instance, NDT [5] and other sensors [6]), the number of applications that can be automated has been extended [7,8]. However, in some scenarios, a single continuum manipulator is insufficient to complete a complicated task [9]. Take, for example, the need to cut and retrieve a solid sample from a confined space for decommissioning planning [8]. At least two continuum arms working collaboratively are needed to perform the task (i.e., one to hold and one to remove the sample). However, correctly operating a continuum manipulator-based robot, which is likely to have many DoF’s, involves calculating the inverse kinematics and/or dynamics [10], which is complicated for a single continuum arm conducting a straightforward task, let alone for multiple continuum arms [11].

Significant effort has been expended recently in developing continuum robots to cope with the demands of specific challenging environments. For example, a particularly long and slender continuum arm (length: 1.5 m; diameter: 12.5 mm) was developed for aeroengine maintenance [7]. Further, a 3 m long, dual segment (a dexterous section supported by a rigid section) continuum robot with 60 joints was designed to facilitate the maintenance of a nuclear power plant [12]. Most of these robots are single manipulators designed to operate alone. However, there have been several attempts to develop continuum robots that either work together with other robots or have multiple manipulators that operate collaboratively. For instance, two identical robotic systems with different end effectors (one with a camera and flame ignitor, another with a thermal spray nozzle), developed at Nottingham University in collaboration with Rolls Royce plc [13], were deployed to repair the coatings on the inside of a very cramped aeroengine collaboratively from two opposite inspection holes [12]. Dual-arm continuum robots have been considered for telesurgery to perform ‘two-handed’ tasks, including suture passing and knot tying, such as the system developed by Columbia University and Vanderbilt University, where two or more concentric tube continuum arms extend from a rigid base [13,14]. As the shape of the arms here is dictated by complex anatomical features, the necessarily high-accuracy control relies on a complicated kinetostatic model [15]. A hybrid dual-arm robotic system, developed by a consortium of universities in China, consists of a combination of a rigid manipulator and a continuum manipulator, which have been designed with the intent to operate collaboratively [16]. However, to the best knowledge of the authors to date, the proposed design has been simulated but not yet validated experimentally.

As these robots have been largely used in challenging environments that humans cannot access (e.g., in dangerous conditions [17] and/or in confined spaces [18]) to perform tasks that often mimic or extend human performance, that is, multiple arms with complex end effectors working together, different collaboration strategies (e.g., robot–robot [19,20] and human–robot [21,22]) have been developed. For example, six commercial robots have been deployed to work collaboratively in a large-scale additive manufacturing process to reduce manufacturing time and improve flexibility [23]. The activities of these robots are coordinated by a central control system that uses process sequencing to ensure that robot–robot interference is avoided. In many scenarios, especially in manufacturing and assembly, humans have also been regarded as part of the robotic system. This is called human–robot collaboration, where humans and robots carry out coordinated activities to achieve common goals, for example, the assembly of a pump in a hybrid production cell [24] wherein tasks are allocated between humans and robots depending on complexity and operation time. This often requires interactions between human and robot so that intent is understood and safety is assured. That was achieved through the monitoring of body gestures in that were interpreted as commands to the robot. Capturing intent by monitoring the motion of both humans and robots and incorporating it into an adaptive impedance control to enable close collaboration was also considered in [25].

Even though a lot of effort has been spent on the control of multiple collaboratively operating robots, a number of challenges and solution gaps exist, especially for continuum robots. These include: (1) a lack of an analytical solution for continuum robots comprising multiple sections, causing difficulty for real-time control; (2) continuum robots with many redundant actuators, having multiple solutions (i.e., configurations or shape) for the desired tip orientation; (3) the continuous adaption of the shape of the continuum robot for a given toolpath being important to avoid vibration and collision.

To address these challenges for cable-driven collaborative continuum robots, this paper focuses on the following aspects: First, the modelling of a dual 6-DoF continuum arm robot to establish the relationship between the length of the driving cables and the pose of the end effectors, as well as determining and analysing the collaboration workspace, as shown in Section 3; an optimization-based path planning algorithm to obtain the continuous shape of the continuum arms using the developed inverse kinematic model, which is evaluated using case studies in Section 4; and an experimental demonstration of the algorithm for collaborative operation in Section 5 using the robot prototype shown in the next section. Finally, the conclusions are presented in Section 6.

## 2. Design of a Dual-Arm Continuum Robot

In this paper, a novel dual-arm robotic system is proposed, wherein two 6-DoF continuum, or snake, arms have been incorporated to perform cooperative tasks using specialized end effectors (i.e., as an example here, a rotary grinder is used on the end of the left arm, with a gripper on the right).

The concept of dual-arm operation and structure of the arm originates from the biomimetic structures (e.g., dual-arm from human beings, arm from snakes). Each arm has 6-DoF to enable the dexterity operation capability. When the two arms work collaboratively with their end effectors (e.g., gripper and grinder), it resembles with human beings in using a fork and knife by the arms. To obtain the dexterity compatibility of the arm, the modular 2-DoF sections were serially connected and actuated by the cables, which is inspired by the working principle of snakes and arms of human beings. For each 2-DoF section, the modular disks were connected by ball joints (spine structure) and flexible hinges (muscles around the spine), which are mimicked by the body structure of snakes, enabling the system to have good mechanical performance. With the aforementioned bio-inspired structure, a new kind of dual-arm robotic system was developed for coping with the task in remote challenging environments.

This setup has been configured to operate as a solid sample removal and retrieval system mounted on an underwater remotely operated vehicle (ROV) in a decommissioning scenario as an example. However, it is easily envisioned how it could be adapted for a range of remote engineering and manufacturing applications. In the scenario mentioned, the gripper can hold the structure from which a sample is required, while the grinder on the other arm can be controlled to follow the planned cutting path to remove the sample from the structure. Then the ROV can take the obtained sample, as well as the continuum robot system back to a collection point. The system setup is shown in Figure 1.

Each biomimetic snake arm is composed of three 2-DoF sections, each providing 6-DoF for each arm. This allows for the control of the position and orientation of the end effectors. These arms are designed to work together collaboratively and have the same length (540 mm) and a small diameter (40 mm), giving a slenderness ratio of 0.074.

To improve the stiffness characteristics of the continuum arms and increase the permissible payload and accuracy, each section is composed of an array of 2-DoF segments connected in series, each formed of three parallel flexure hinges between two rigid disks guided by a central ball joint [8]. In each section, the diameter of the disks and the width of the flexure hinges are used to optimize the stiffness, with Section 3 being stiffer than Section 2 (as Section 3 has to support the most load), which in turn is stiffer than Section 1 (as Section 1 is ideally the most dexterous section). Cable-driven actuation (shown in detail in Figure 2) is used to provide position and force control of the arms and, in turn, the end effectors. Each 2-DoF section is actuated by three cables, and therefore, 18 cables and their associated motor units are needed to actuate the dual arms [8,26,27]. The gripper end effector is also cable-actuated, and so another motor unit is required, bringing the number of motor units to be controlled to 19. The grinder end effector is motorized but has its own direct drive system that transmits torque to the cutting tool by way of a long flexible drive shaft.

## 3. Modelling of the Continuum Robot

Unlike conventional rigid robots, controlling this kind of continuum robots requires adjusting the length of the driving cables. However, as there are normally more than two driving cables required to control each 2-DoF section, the redundancy in actuation will cause challenges in kinematic modelling and control. Further, as each 6-DoF continuum arm is formed by connecting three 2-DoF sections in series, the analytical solution for the inverse kinematics (i.e., given the position and orientation of the end effector, solving the shape parameters of the continuum arm) is difficult.

### 3.1. Kinematic Model

To describe the spatial configuration of the 6-DoF continuum arm, the piecewise constant-curvature theory (PCCT) is used in this paper [28,29]. Based on this assumption, the curvature of each 2-DoF section is regarded as part of a circle with a specific radius. For the forward kinematics, the position and orientation of the tip of the 6-DoF continuum arm can be determined given the bending and phase angles of the three sections. Then, given the shape of the 6-DoF continuum arm, the change in length of the driving cables can be determined as well. For the inverse kinematics, it is necessary to determine the bending and phase angles (θ and φ, respectively, in Figure 3a) of the three sections to satisfy the required position and orientation of the end effector. As the equations are transcendental in nature, there is no analytical solution for the inverse kinematics, and so a novel optimization-based method has been developed to solve this problem.

The coordinate systems for the three sections that make up each 6-DoF arm are shown in Figure 3. A local coordinate system ({$O_{i}$}, *i* = 1,2,3) is attached to the base of each 2-DoF continuum section to define the bending and phase angles to parameterize the 2-DoF of each section. One coordinate system {$O_{3}$} is attached to the base of the end effector to define its position and orientation during operation.

Consider a single 2-DoF continuum section, which consists of a series of identical structured segments, as an example. Based on the PCCT, the bending along its axis is assumed to cause a constant curvature, which means that the bending angle for the *j*-th segment in the *i*-th section, $\theta_{ij}$, can be given as:(1)$$\theta_{ij}=\frac{\theta_{i}}{n}$$
where $\theta_{i}$ is the bending angle, and n is the number of segments in the *i*-th section.

The transformation matrix from {$O_{ij}$} to {$O_{i,j+1}$} can be defined as follows: first, {$O_{ij}$} moves from its origin $O_{ij}$ to $O_{i,j+1}$ along the $Z_{ij}$ axis; then {$O_{ij}$} rotates around the $Y_{i,j+1}$ axis by angle $\theta_{ij}$. Thus, the homogeneous transformation matrix can be expressed as:(2)T Oi,jOi,j+1=RXθi,jDi,j01
where $R_{X}\left(\theta_{i,j}\right)$ is the rotation matrix generated by rotating about the $Y_{i,j+1}$ axis by angle $\theta_{ij}$ of {$O_{i,j+1}$}; $D_{i,j}$ is the relative position vector from $O_{ij}$ to $O_{i,j+1}$ in the {$O_{i,j}$} frame.

By postmultiplying the transformation matrices of the three sections, the entire kinematic model of the continuum robot can be established:(3)T O1,0O3,n3−1=T0⋅∏j=0n1−1T O1,jO1,j+1⋅∏j=0n2−1T O2,jO2,j+1⋅∏j=0n3−1T O3,jO3,j+1
where $n_{1}$, $n_{2}$, and $n_{3}$ are the number of segments in the three continuum sections, respectively; $T_{0}$ is the initial transformation matrix of the continuum arm, which can be written as:(4)$$T_{0}=\left[\begin{array}{cc} R_{0} & P_{0}\\ 0 & 1 \end{array}\right]$$
where $R_{0}$ and $P_{0}$ are the initial rotation matrix and position vector, which defines the initial orientation and position of the continuum arm.

The shape of the 6-DoF continuum arm depends on the length of the driving cables. Taking an arbitrary segment as an example, the closed-loop vector can be expressed as:(5)$$l_{i,j}=t+R\left(s+p_{i}\right)-b_{i}$$
where $l_{i,\;j}$ is the relative position vector of the ends of the *i*-th driving cable in the *j*-th segment; t is the position vector of the centre of the ball joint in {$O_{B}$}; s is the position vector of the centre of the upper platform in {$O_{U}$}; $p_{i}$ is the position vector from the upper platform centre, $U_{c}$, to the cable attaching point in the upper platform, $U_{i}$; $b_{i}$ is the position vector of the cable attaching point in the lower platform, $B_{i}$; and R is the rotation matrix of the upper platform. Please refer to [8] for details.

The change in length of the *i*-th driving cable in the *j*-th segment can be expressed by taking the norm of the vector $l_{i,j}$:(6)$$l_{i,j}=\left|l_{i,j}\right|$$

Due to the PCCT assumption, wherein the curvature and, hence, the change in length of the *i*-th driving cable are the same for all segments within a section, the total change in length of the *i*-th driving cable for the whole section (total of all segments) can be expressed as:(7)$$l_{i}=nl_{i,j}$$
where n is the number of 2-DoF segments in each section.

These calculations form the forward kinematic equations for the 6-DoF continuum arm. For the inverse kinematic calculations, the shape parameters (i.e., bending and phase angles) of the three sections need to be calculated for a given position and orientation of the end effector. Then, by using the kinematic model developed in this section, the calculated change in length of the driving cables can be used to control the actuation system.

### 3.2. Cooperative Workspace

Here, the robot consists of two 6-DoF continuum arms, comparable to a human with two dexterous arms working together to complete complicated tasks. The calculation of the workspace for a single 6-DoF continuum robot can be conducted using the kinematic models developed in the last section. However, for the workspace of the dual-arm continuum robot working collaboratively on a task, it is more complicated as the workspace not only depends on the permissible configurations (e.g., bending stroke and length of each 2-DoF segment) of each 6-DoF continuum arm, but also will be dependent on the dimensions and shape of any workpiece that the arms are both working on.

Following the biomimetic design of the continuum arms, the structural dimensions (e.g., diameter of the continuum arm and width of the flexure hinge) in Section 3 are designed to be larger than Section 2 and, in turn, Section 1 to improve the stiffness of the arm. The details of the structural parameters for a 6-DoF continuum arm can be seen in Table 1.

Based on the kinematic model developed in Section 3.1, the workspace of the dual-arm continuum robot can be calculated by plotting the tip position of the two 6-DoF continuum arms when changing the bending and phase angles within their strokes (see Figure 4).

It can be seen from Figure 4a that the workspace of each 6-DoF continuum arm alone is a segment of a sphere owing to symmetry. For the cooperation workspace, it will be much dependent on the task and size of the object. For the collaborative task of carrying a large-sized object, the collaborative workspace will be larger than the overlapping workspace. Meanwhile, for the collaborative task of sample removal, the collaborative workspace will be much similar to the overlapping workspace. For the overlapping workspace, the tip of both 6-DoF continuum arms can occupy any point within this area, enabling the completion of the designed task (i.e., sample retrieval). Figure 4b shows a section view of the workspace of the dual-arm continuum robot (defined by the yellow curve in the shape of a cownose ray) and can be used to plan the path of the two continuum arms working collaboratively.

## 4. Modelling of the Continuum Robot

To complete a task collaboratively, the paths of the two 6-DoF continuum arms, and specifically the end effectors, need to be planned based on their specific functions. However, as there is no analytical solution for the inverse kinematics, the shape parameters (i.e., bending and phase angles) of each of the 2-DoF segments in each arm cannot be obtained directly, requiring a new optimization-based strategy.

### 4.1. Formulation of the Objective Function

In the remote sample removal case study, the two continuum arms need to work collaboratively. To conduct the operation, the continuum arm with the gripper as an end effector holds the structure at a convenient location to secure the robot and the sample (such as the corner, as this can simplify the cutting path of the grinder), while the other arm with the grinder at the end moves along the specified cutting path to remove the sample from the structure. As the dual-arm robotic system will be attached to a commercial ROV, the initial positioning of the gripper can be aided by the operation of the ROV (see Figure 5). However, the path of the other arm needs to be carefully planned to complete the task.

To successfully remove the sample from the structure, the following challenges need to be considered during path planning: (1) the path should fit entirely within the workspace of the arm; (2) the arm should not collide with other features (e.g., the other arm); and (3) the approach of the high-speed cutting tool, or grinder, towards the structure needs to be taken with care to enable effective cutting (see enlarged view of Figure 5).

As the analytical solution for the inverse kinematics is hard to obtain, a new optimization-based approach is proposed to obtain the shape parameters (e.g., bending and phase angles) for each of the three 2-DoF sections of each arm under the specified position and orientation constraints. Taking a single arm as an example, the optimization procedure can be described thus:(1)Generate the path of the end effector, which should fit entirely within the workspace.(2)Parameterize the path at discrete points.(3)Define the optimization objective function based on the position and orientation constraints of the end effector at each point along the path.(4)Run the optimization function to find the shape parameters of the continuum arm to meet the given constraints.(5)Repeat the optimization procedure to all the points to obtain the shape parameters of the continuum arm.

The optimization objective function of the arm can be written as:(8)$$f\left(\begin{array}{ccc} \theta_{0}+\delta_{\theta } & \phi_{0}+\delta_{\phi } & d_{0}+\phi \delta_{d}\; \end{array}\right)=max\left(k_{1}\Delta Tip_{\mathrm{Position}}+k_{2}\Delta Tip_{\mathrm{Orientation}}\right)$$
where $\delta_{\theta }$, $\delta_{\phi }$, and $\delta_{d}$ are specified increments of the bending angle, phase angle, and tip advancing distance, respectively; max is the maximization function; $\Delta Tip_{\mathrm{Position}}$ and $\Delta Tip_{\mathrm{Orientation}}$ are the position and orientation errors (i.e., between the actual and planned values) of the tip; and $k_{1}$ and $k_{2}$ are the coefficients to assign the weights to the position and orientation subobjectives for the overall objective function.

Specially, the position and orientation errors of the tip can be expressed as:(9)$$\Delta Tip_{\mathrm{Position},i}=norm\left(P_{\mathrm{Tip}}\left(\begin{array}{ccc} \theta_{0}+\delta_{\theta } & \phi_{0}+\delta_{\phi } & d_{0}+\phi \delta_{d}\; \end{array}\right)-P_{\mathrm{Toolpath},i}\right)$$
(10)$$\Delta Tip_{\mathrm{Orientation},i}={\left[\log\left(R_{\mathrm{Tip}\left(\begin{array}{ccc} \theta_{0}+\delta_{\theta } & \phi_{0}+\delta_{\phi } & d_{0}+\phi \delta_{d}\; \end{array}\right)}^{T}R_{\mathrm{Toolpath},i}\right)\right]}^{\vee }$$
where norm is the norm length of the vector; log is the matrix natural logarithm, ∨ is the operator that maps change in orientation from so(3) to R3, $P_{\mathrm{Tip}}\left(\begin{array}{ccc} \theta_{0}+\delta_{\theta } & \phi_{0}+\delta_{\phi } & d_{0}+\phi \delta_{d}\; \end{array}\right)$ is the tip position at the new point, $P_{\mathrm{Toolpath},i}$ is the planned position of the *i*-th point, $R_{\mathrm{Tip}\left(\theta_{0}+\delta \theta ,\;\;\phi_{0}+\delta \phi ,\;\;d_{0}+\delta d\right)}$ is the orientation of the tip at the new point, and $R_{\mathrm{Toolpath},i}$ is the planned orientation of the *i*-th point.

An example path for the arm with the grinder end effector in the remote sample removal case study is shown in Figure 6. The full path composed of the approaching and leaving stages (in pink) and operating stage (in blue) fits within the workspace of the 6-DoF continuum arm. The configuration of the 6-DoF continuum arm (configuration: bending and phase angles for the three sections are $\theta_{1},\;\theta_{2}$ and $\theta_{3}$, $\varphi_{1},\;\varphi_{2}$ and $\varphi_{3}$, respectively) at two representative points along the path are shown (Section 1: black, Section 2: green, and Section 3: red) in Figure 6a.

The definitions of the tip position and orientation error used in the subobjective functions are shown in Figure 6a,b. The position error is defined to be the shortest distance between the actual position and the planned position, which can be expressed as $\sqrt{\Delta x^{2}+\Delta y^{2}+\Delta z^{2}}$ (see Figure 6a). Similarly, the orientation error is defined to be the difference in orientation of the three axes, which can be calculated by using Equation (10) (see in Figure 6b).

### 4.2. Toolpath Generation Algorithm

As optimization is a time-consuming procedure, it is unrealistic to infinitely discretize the planned path, even though this could improve the cutting efficacy of the system. To solve this problem, it is proposed that the planned path is discretized into a relatively small number of points whereat the optimization algorithm is used to calculate the configuration of the arms, and then interpolation is used to obtain the results at intermediate locations.

Taking the cutting operation as an example, the full process includes the following steps: (1) the end effector approaches the structure from its initial position at a high speed; (2) the end effector moves along the planned cutting path to remove the sample from the structure at a slow speed; (3) the end effector leaves the structure along the leaving path at a high speed; and finally, (4) the end effector returns to the initial position along the return path at a high speed (see Figure 7a).

In order to maintain cutting efficiency, the position and orientation of the cutting tool have additional constraints. The spindle axis should be kept normal to the structure surface, and the axis should be coincident with the planned point, with the location of the tip consistent with the required cutting depth. Thus, the following equations should be satisfied:(11)$${\Delta \mathrm{Position}}_{\mathrm{Toolpath},i}=\min\left(\left(P_{\mathrm{Tip},i}-P_{\mathrm{Toolpath},i}\right)\right)\;{\Delta \mathrm{Orientation}}_{\mathrm{Toolpath},i}=\min\left({\left[\log\left(R_{\mathrm{Tip},i}^{T}R_{\mathrm{Toolpath},i}\right)\right]}^{\vee }\right)$$
where, $\Delta {\mathrm{Position}}_{\mathrm{Toolpath},i}$ and $\Delta {\mathrm{Orientation}}_{\mathrm{Toolpath},i}$ are the position and orientation errors of the end effector, and $P_{\mathrm{Tip},i}$ and $R_{\mathrm{Tip},i}$ are the specified position vector and orientation matrix of the end effector.

Given this procedure, the bending angle ($\theta_{i,j}$) and phase angle ($\varphi_{i,j}$) of the *j*-th section of each arm at the *i*-th point along the toolpath can be determined. Similarly, the bending angle ($\theta_{i+1,j}$) and phase angle ($\varphi_{i+1,j}$) at the adjacent (*i* + 1-th) point can be obtained in the same way. The bending angle ($\theta_{i,j,m}$) and phase angle ($\varphi_{i,j,m}$) at intermediate points can be calculated via interpolation by:(12)$$\left\{\begin{array}{c} \theta_{i,j,m}=\theta_{i,j}+m\frac{\theta_{i+1,j}-\theta_{i,j}}{n}\\ \varphi_{i,j,m}=\varphi_{i,j}+m\frac{\varphi_{i+1,j}-\varphi_{i,j}}{n} \end{array}\right.,\;m=1,2,\ldots ,n;\;n\geq 2$$
where n is the number of intermediate points between the adjacent toolpath points (i.e., *i*-th and *i* + 1-th, respectively). This is an efficient way for calculating the shape parameters of the 6-DoF continuum arms.

### 4.3. Simulation-Based Evaluation

This optimization-based inverse kinematics algorithm allows the shape parameters of the dual-arm continuum robot to be calculated for a given toolpath for any application. As the two 6-DoF continuum arms are similar, only one is used to demonstrate the proposed algorithm.

The scenario is as defined in Figure 7, where the toolpath is located between the two continuum arms and c.a. 450 mm distance away from the actuation system. So that the end effector follows the desired toolpath while maintaining the position of the tip, the shape of the three sections of the arm needs to be adjusted dynamically. As such, the toolpath is parameterized into discrete points, and the algorithm run at each point to find the suitable set of bending and phase angles that fulfil the convergence conditions of the algorithm.

The change in shape of the arm for the given toolpath is shown in Figure 8. In the case studies of this paper, the constraint for the optimization is the maximum bending angle of the three sections, which is set based on the physical structure design of the 2-DoF sections. By running the developed algorithm of this paper and using the commercial build-in optimization algorithm of the MATLAB software (i.e., fmincon nonlinear optimization), the shape variation of the continuum arm can be obtained.

It can be seen in Figure 8 that the shape of the three sections (defined by the bending angle, $\theta_{i}$, and phase angle, $\varphi_{i}$) can vary significantly while trying to keep the tip of the arm at the intended position. As there are no constraints on the shape, the change in shape at adjacent steps is not continuous, which means that there are large changes. Taking Figure 8c as an example, even though adjacent points (e.g., i and i+1) are close, the two configurations may be in different quadrants. To solve this problem, the change in angle should be limited, as demonstrated below.

To provide the kinematic parameters for controlling the 6-DoF continuum arm, the shape parameters (i.e., bending angle and phase angle) of the three sections, as well as the error between the calculated path and the desired path, were obtained and are displayed in Figure 9.

It can be seen from Figure 9a that the bending angle and phase angle of the three sections change to control the end effector position. The bending angle is limited in the optimization algorithm to stay in the range [0°, 90°], which is observed for each section. The phase angles are not constrained, and while they stay within the same quadrant most of the time as the end effector moves from point to point, occasionally, the phase angle jumps to another quadrant. This is bad for the performance of the system. Taking the phase angle of Section 2 as an example (green dotted line), the angle jumps into a different quadrant at step 4 and returns to the initial quadrant at step 6. The main reason is that there are multiple solutions for the shape configuration of the continuum arm for any given tip position due to the symmetrical structure of the arm. It can be seen from Figure 9b that high accuracy is still achieved; that is, the position errors between the planned toolpath and desired toolpath are within 0.3 μm, despite of these jumps in shape.

As having multiple solutions for the shape configuration of the arm has implications for the control and stability of the system, this was investigated further. Taking the scenario shown in Figure 8 as an example, additional constraints were applied to the phase angles of the three sections in the optimization algorithm. Here, the phase angles of the three sections were kept constant at 0° or 180°, which means that the continuum arm can only bend within the XOZ plane. Figure 10 shows the change in shape of the 6-DoF continuum arm while tracking the desired toolpath with the additional constraints of the phase angles. It can be seen that the configuration of the arm gradually and smoothly changes as the end effector moves along the given path, which demonstrates the possibility of regulating the phase angle to improve the performance of the system.

Under the given constraints on the phase angles of the 6-DoF continuum arm, the bending angles changed gradually while the end effector moved along the path, as seen in Figure 11a. As can be seen, the phase angles of the three sections are either 0° or 180°, and this keeps the continuum arm within a plane. The bending varies within the range [0° 90°], as before. Figure 11b shows the error between the calculated end effector position and the desired location at each point. It can be seen that the additional constraints resulted in a higher accuracy (within 0.15 μm) than without the constraints (within 0.3 μm).

## 5. Experimental Results Evaluation

In this section, the optimization-based inverse kinematic algorithm developed above will be employed in the control of a prototype continuum dual-arm robot. Here, the two 6-DoF continuum arms perform a task collaboratively, specifically the remote sample removal operation described previously. To achieve this, the shape parameters (i.e., the phase and bending angles) of each section of the arms were calculated using the optimization-based inverse kinematic algorithm at each point on the planned toolpath and used as inputs in the control system to regulate the shape of the two continuum arms.

### 5.1. Experimental Setup

The experimental setup of the dual-arm continuum robot is shown in Figure 12a, which is composed of two 6-DoF continuum arms, each with a specific end effector (i.e., grinder and gripper, respectively), actuation system (i.e., 18 motors for the two continuum arms and 2 motors for the end effectors; note: all the motors are equipped with their own independent real-time closed-loop controllers), power system (battery), GUI system, and communication system to a remote computer. As before, the continuum arms and gripper end effector are cable-actuated. The length of each cable is adjusted by controlling the angular displacement of a dedicated motor unit that includes an encoder, which provides the data for a displacement-based closed-loop controller. The grinder end effector (seen in Figure 12c) is controlled by a speed-based closed-loop controller to regulate the cutting speed. To achieve closed-loop control of the motors at high speed, two National Instruments sbRIO-9627 FPGA–based embedded boards were used (seen in Figure 12b). A router is used to build the local area network for the bidirectional communication between the computer and FPGAs.

The planning strategy follows that defined in Section 4. The robot is positioned near the structure, and the path that the gripper will follow is set (including the approach, cutting path, exit path, and return), and both paths are discretized into a set of points. The optimization-based inverse kinematic algorithm is then used to calculate the required phase and bending angles for each section of the arms at each point. The shape of the arms at intermediate locations is determined using interpolation. The piecewise constant-curvature (PCC) theory is used to calculate the necessary change in the length of the driving cables. During operation, the closed-loop controllers dynamically adjust the rotation of each motor to change the length of the driving cables to the desired value. At the same time, the spindle speed of the grinder is adjusted, and the gripper is actuated, depending on their respective positions along their paths. The entire workflow of the collaborative operation can be seen in Figure 13.

### 5.2. Collaborative Operation Demonstration

The paths that the gripper and grinder end effectors follow during the remote sample removal case study are shown in Figure 14. The full operation of the dual-arm continuum robot can be divided into the following stages:(1)Initial configuration of the arms (which defines the initial bending and phase angles of each section), ensuring that the arms have sufficient space and reach to perform the task;(2)The clamping stage, where arm with the gripper moves towards the structure and holds the edge at an appropriate position;(3)The milling stage, where another arm moves along the planned paths (approach, cutting, exit, and return) to remove the sample from the structure;(4)The returning path of the gripper, where the removed sample is moved by the gripper to ease the return of the grinder.(5)The returning path of the grinder, where the grinder returns to its initial configuration.

In an extension to this scenario, the dual-arm robotic system can be mounted on a locomotive device (e.g., an ROV for operating in underwater environments) for the transportation of the sample for analysis.

After the paths of the two end effectors have been determined, the optimization-based inverse kinematic algorithm developed in Section 4 can be used to calculate the required change in the shape of the two 6-DoF continuum arms, as seen in Figure 15. For visibility, the evolution of the shape of the arms is displayed in stages. The shape of the arm with the grinder during the cutting stage is displayed in Figure 15a, while the shape during the full path is shown in Figure 15b. Similarly, for the arm with the gripper, the shape evolution during the collection stage is displayed in (c), and the full path in (d). To show their spatial relationships and emphasize the collaborative nature of this operation, the changes in shape of both continuum arms were plotted together in (e). Recall that the axis of the spindle of the grinder should be kept normal to the surface of the structure for efficient cutting. To demonstrate that the developed algorithm can ensure this, a section view of (e) is displayed within the enlarged view. It can be seen that the axis of the spindle is indeed maintained correctly during the cutting operation, which demonstrates the advantage of the developed algorithm.

This scenario was investigated experimentally using the setup shown in Figure 16a. To simulate the real scenario, a structure composed of a foam plate was fixed to the bench by a clamp near the robot. The path, and the shape of the two 6-DoF continuum arms at each point along that path, was determined as shown in Figure 15. The control strategy followed that which is described earlier in this section.

Figure 16b shows the experimental results for completing the collaborative task described above, in which the red and blue dotted curves are the planned and real toolpaths, respectively. Overall, the control of the end effectors was achieved with the proposed closed-loop controller and inverse kinematic algorithm. In this example, the average error between the planned and real toolpaths was within 2.5 mm, which is sufficient for low-accuracy applications, such as sample retrieval. The ability of the dual-arm collaborative robot to operate collaboratively and be controlled using the optimization-based inverse kinematic algorithm has also been demonstrated. Note that the functionality and reach of this robot can be extended through integration with a mobile platform, such as an ROV, so that it can operate in inaccessible and /or hazardous areas, like underwater or in confined spaces.

## 6. Conclusions

In this paper, a dual-arm continuum robot (wherein each 6-DoF arm is formed from three cable-driven 2-DoF sections, each composed of a series of modular 2-DoF segments) has been developed to perform cooperative tasks remotely. The control of the collaborative arms was achieved using a new optimization-based inverse kinematic algorithm. This is used to determine the required bending and phase angles of each section of the continuum arms and incorporate the geometrical constraints on the arms and operation requirements of the end effectors. This algorithm was augmented by a novel strategy for generating the operational sequences and associated paths of the two 6-DoF continuum arms. The utility of the robot and control system for collaborative tasks was demonstrated by a case study wherein the robot was used to remove a sample from a structure remotely. This scenario was investigated numerically and experimentally, and it was shown that the system and control system can conduct collaborative tasks effectively. The accuracy was sufficient for tasks such as sample removal and can be improved through additional feedback of the end effector position and orientation, perhaps using image analysis. The robot has been designed to be integrated to a mobile platform, which will further extend the reach and functionality for use in confined and hazardous areas.

## Figures and Tables

**Figure 1 biomimetics-08-00004-f001:**
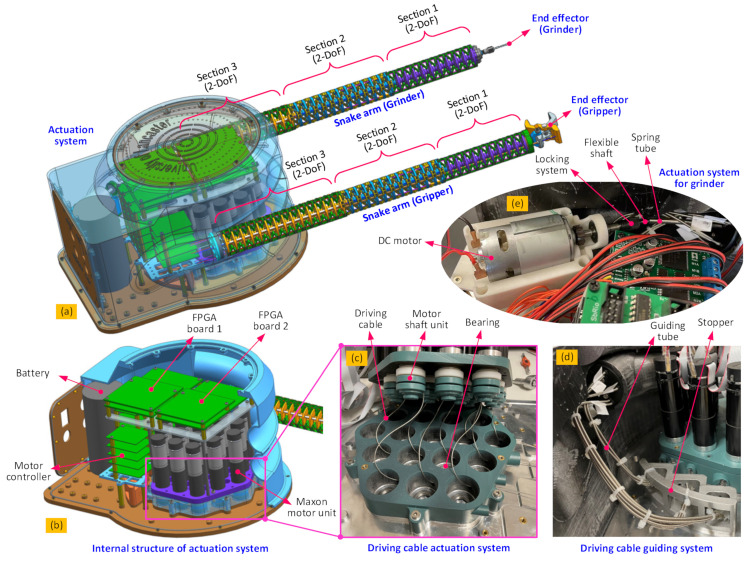
Detailed structural schematic of the dual-arm continuum robot: (**a**) is the overall structure of the continuum robot with the extended dual continuum arms, (**b**) is the internal structure of the actuation pack, (**c**) is the actuation principle for the continuum arms [8,26,27], (**d**) is the cable guiding system for controlling the continuum arm and gripper, (**e**) is the actuation system for the grinder.

**Figure 2 biomimetics-08-00004-f002:**
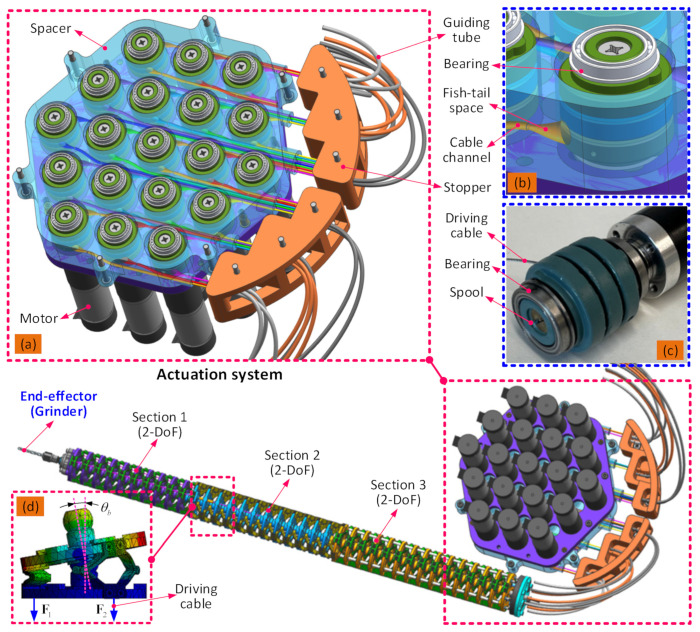
Detailed structural schematic of the dual-arm continuum robot: (**a**) is the overall structure of the continuum robot with the extended dual continuum arms, (**b**) is the internal structure of the actuation pack, (**c**) is the actuation principle for the continuum arms [8,26,27], (**d**) is the cable guiding system for controlling the continuum arm and gripper.

**Figure 3 biomimetics-08-00004-f003:**
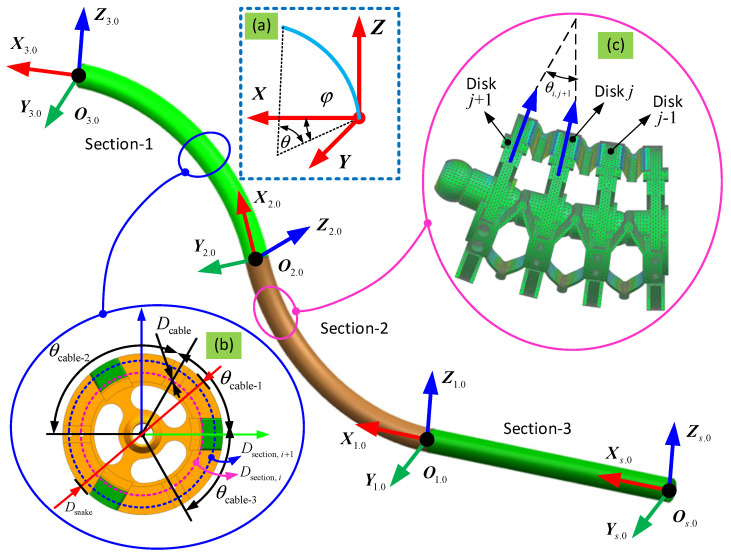
Kinematic definition of the continuum robot formed by three sections: (**a**) definition of the bending and phase angles, which are parameters for the constant curvature bending model of each section; (**b**) section view of a typical disk that makes up each segment; (**c**) kinematic relationship of adjacent disks within each section.

**Figure 4 biomimetics-08-00004-f004:**
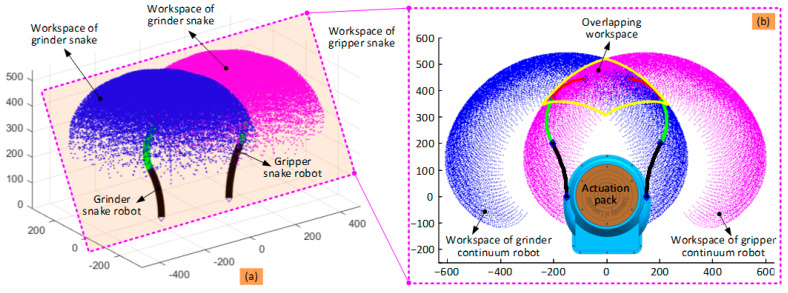
Cooperative workspace of the dual-arm continuum robot: (**a**) is the overall view of the workspace for the two continuum robots; (**b**) is the section view of the cooperative workspace. Note: for visibility, the bending ranges for the three sections of the two continuum robots in (**a**) are 30°, 45°, and 60°, respectively, while it is 90° in (**b**). The phase angles are within [0°, 360°] in both (**a**,**b**).

**Figure 5 biomimetics-08-00004-f005:**
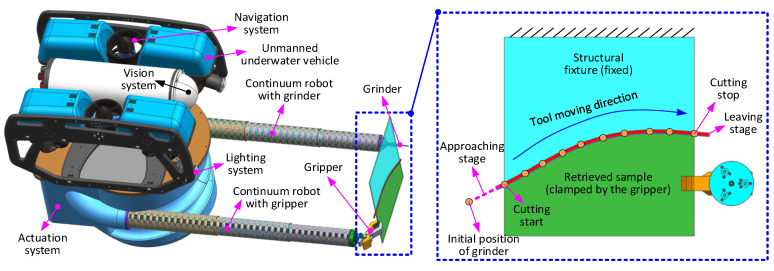
Operating principle of the dual-arm continuum robot in the remote sample removal case study. One arm with the gripper at the end holds the structure, while the other arm with the grinder moves along the planned path to cut the sample from the structure (see the enlarged view).

**Figure 6 biomimetics-08-00004-f006:**
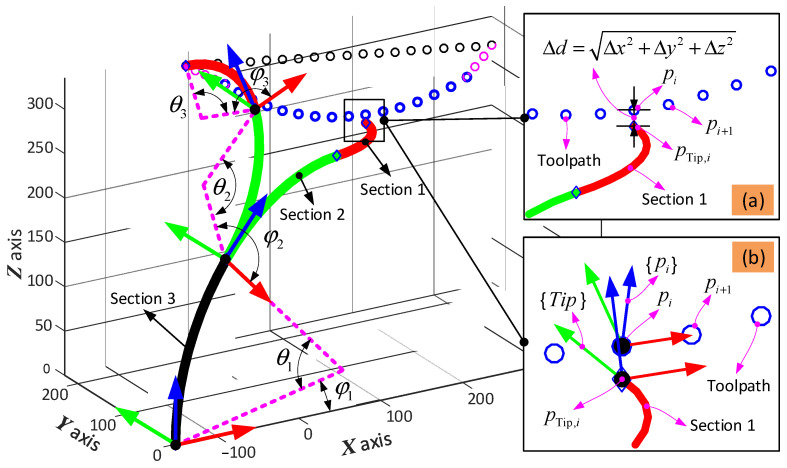
Optimization-based path planning for the 6-DoF continuum arm: (**a**,**b**) are the error definitions of the position and orientation.

**Figure 7 biomimetics-08-00004-f007:**
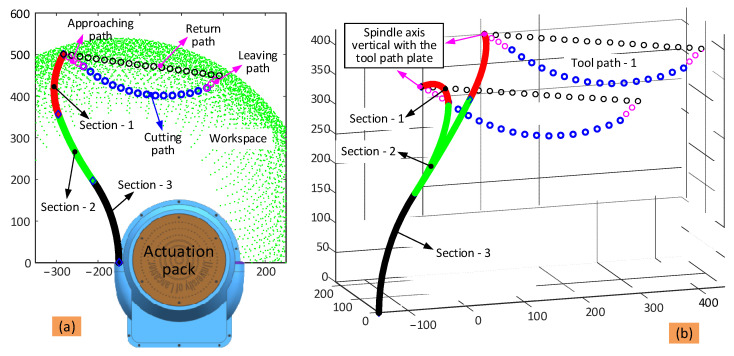
Toolpath planning strategy for the cutting operation in the sample removal case study: (**a**) is the toolpath definition (section view is selected); (**b**) is the overall view of the toolpath to show the configuration of the arm at different relative positions.

**Figure 8 biomimetics-08-00004-f008:**
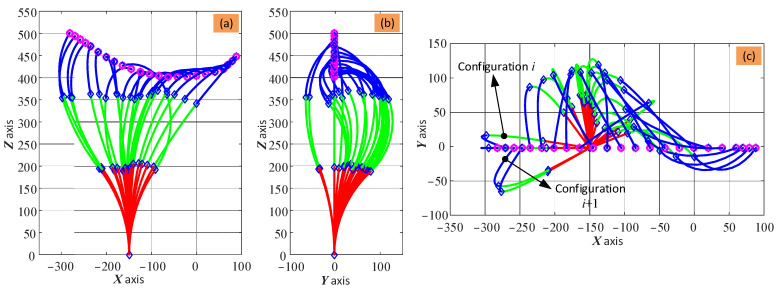
Results of the optimization-based algorithm for the inverse kinematic calculation of the 6-DoF continuum arm at each point along the toolpath: (**a**–**c**) are the front view, side view, and top view of the arm, respectively. Note: there are no constraints for the phase angle ($\varphi_{i},\;i=1,2,3$) of the three sections, and the actuation system is omitted for clarity.

**Figure 9 biomimetics-08-00004-f009:**
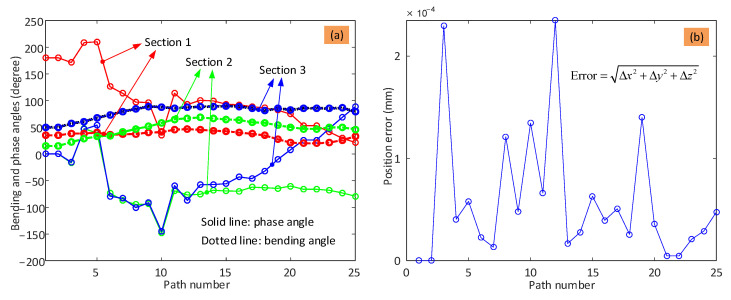
Change in shape parameter for each section of the arm and the optimization accuracy: (**a**) is the variation of the bending angle and phase angle of the three sections; (**b**) is the deviation between the calculated path and desired path. Note: the error is obtained by calculating the distance between the desired path and the planned path (error = $\sqrt{\Delta x^{2}+\Delta y^{2}+\Delta z^{2}}$).

**Figure 10 biomimetics-08-00004-f010:**
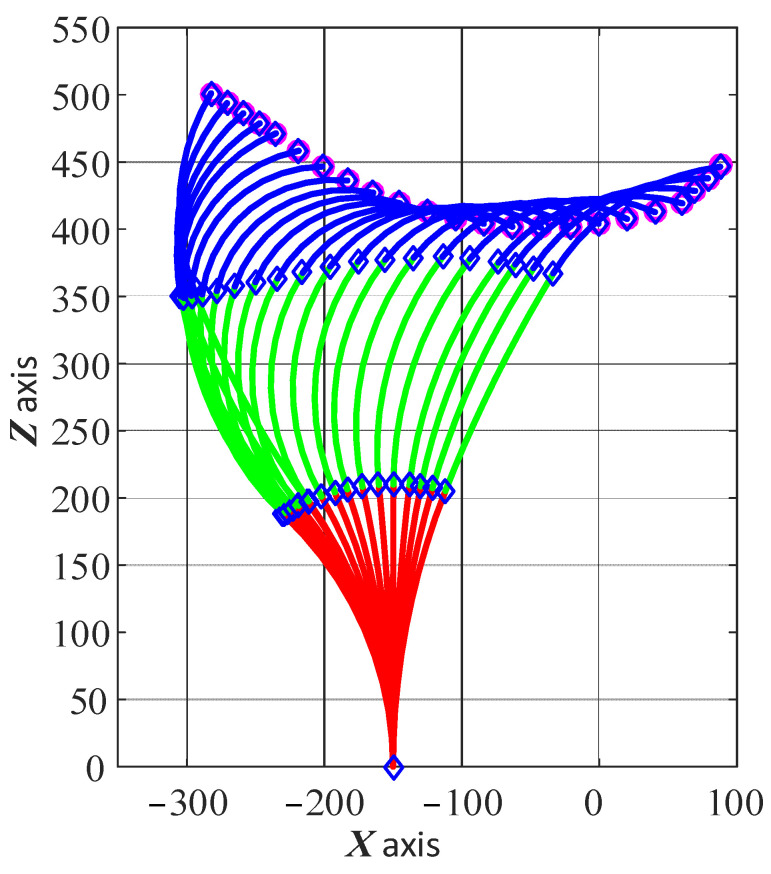
Change in shape of the 6-DoF continuum arm while the end effector moves along the toolpath with the additional constraints of the phase angle.

**Figure 11 biomimetics-08-00004-f011:**
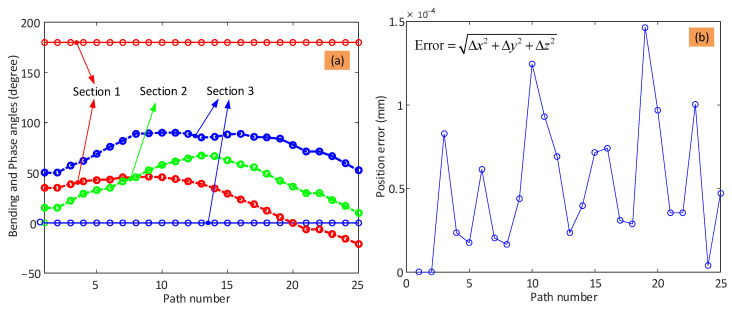
Shape parameters of the 6-DoF continuum arm and the optimization accuracy: (**a**) is the bending angle and phase angle as the end effector moves from point to point; (**b**) is the position error at each point.

**Figure 12 biomimetics-08-00004-f012:**
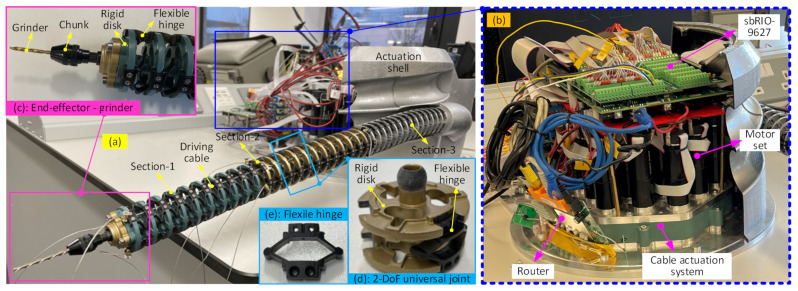
Experimental setup for demonstrating the collaborative operation algorithm: (**a**) is the overall operation setup of the dual-arm continuum robot; (**b**) is the actuation system (i.e., motor units, controllers, and communications); (**c**) shows the arrangement of the cutting tool; (**d**,**e**) are the 2-DoF universal joint and flexible hinge, respectively.

**Figure 13 biomimetics-08-00004-f013:**
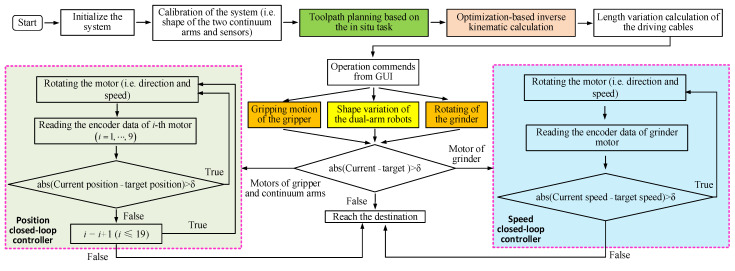
Flowchart of the collaborative operation strategy for the dual-arm continuum robot system.

**Figure 14 biomimetics-08-00004-f014:**
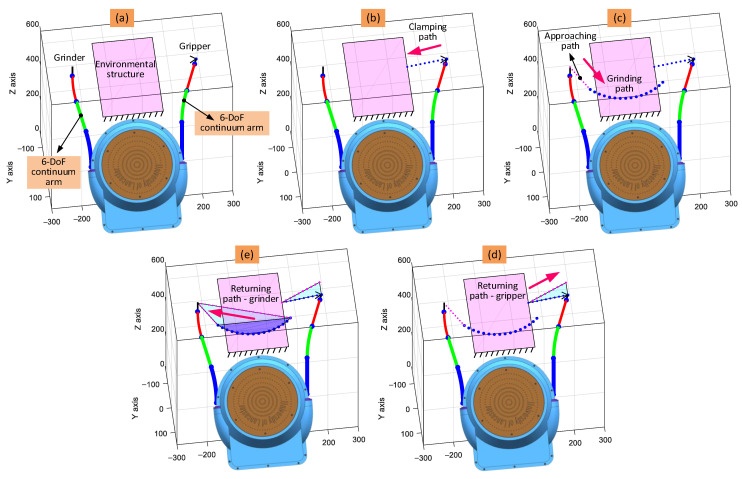
Path planning of the collaborative continuum robots for the remote sample removal from a structure: (**a**) is the initial configuration of the two 6-DoF continuum arms; (**b**,**c**) are the moving path and grinding path of the gripper and grinder continuum arms, respectively; (**d**,**e**) are the returning paths of the gripper and grinder continuum arms, respectively.

**Figure 15 biomimetics-08-00004-f015:**
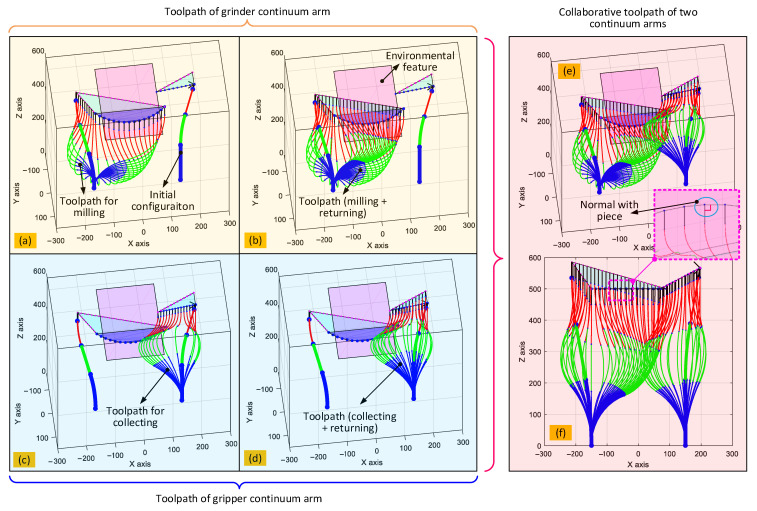
Shape of the collaborative continuum arms during operation calculated proposed optimization-based inverse kinematic algorithm: (**a**,**b**) display the shape of the continuum arm with the grinder, (**c**,**d**) display the shape of the continuum arm with the gripper, (**e**,**f**) show the shape of the two 6-DoF continuum arms working collaboratively. Note: the axis of both end effectors is along the Z axis in this case study.

**Figure 16 biomimetics-08-00004-f016:**
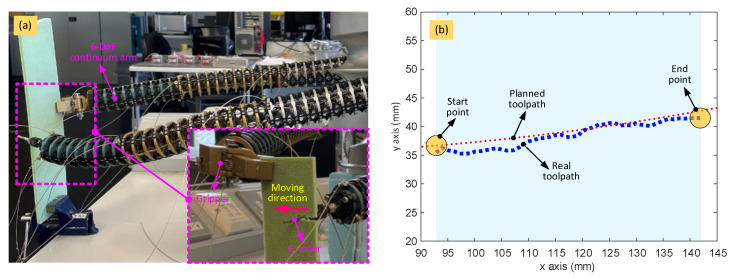
Experimental setup of the collaborative dual-arm robotic system and experimental results: (**a**) snapshot of the system with one arm holding the structure while the other arm conducts the cutting operation for the material removal; (**b**) experimental result between the desired toolpath and the actual toolpath.

**Table 1 biomimetics-08-00004-t001:** Structure parameters of the continuum robot.

Parameters	Length (mm)	Diameter (mm)	Hinge Width (mm)	Hinge Thickness (mm)
Section-1	150	40	4	1
Section-2	180	43	5	1
Section-3	210	46	6	1

## Data Availability

Not applicable.

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
