# Peer review of "Collaborative Continuum Robots for Remote Engineering Operations"

_biomimetics, 2022, doi:10.3390/biomimetics8010004_

Round 1

Reviewer 1 Report

This paper presents the design and operating principles of a dual-arm continuum robot system, which is designed to perform critical tasks within industrial settings. The optimization based inverse kinematic calculation of the 6-DoF continuum arms, and the collaborative operation strategy are discussed in detail. The collaborative principle and algorithms proposed in the paper have been evaluated by a set of experiments to demonstrate the ability of the system to perform in-situ machining operations.

In general, this paper is written in depth and detail, with novel research objects and clear application prospects. The proposed design principle and operation method have reference value. It is recommended to publish the full text.

Reviewer 2 Report

Dear authors, I reviewed your paper entitled collaborative continuum robots for remote engineering operations and I find that the topic is very interesting for the scientific community. The authors present a relevant state of the art, experimental results that could be completed by the characterization of the behavior of an arm or by giving informations in a paragraph "perspectives". Indeed, it would be good to underline the analyses which could be implemented to characterize the mechanical performances of the robot in particular on the aspect of rigidity/stiffness to consider other operations.

It would be good to give some elements before publication:

L110: How are the arms biomimetic?

L159: To parametrize?

L209: What does "complicated task" mean?

Figure 4: the authors talk about cooperative tasks but they depend on the task and/or the size of the object. Maybe it would be good to revise the way you explain it? What do you mean by Tiny L234?

Figure 13: Initial => Initialize ?

L493 : is displayed

L520 : The performance indicator i.e. 2.5mm should appear in the abstract to my point of view
